

**A 2600-yr-long paleoseismic record for the Himalayan Main Frontal Thrust**
**(Western Bhutan)**
Romain Le Roux-Mallouf[1], Matthieu Ferry[2], Rodolphe Cattin[2], Jean-François Ritz[2], Dowchu
Drukpa[2,3], Phuntsho Pelgay[3]
[1]*Geolithe, Research and Development Department, Rue des Becasses, 38920, Crolles*
[2]*Géosciences Montpellier, CNRS, UMR5243, Université de Montpellier, Place E. Bataillon,*
*34095 Montpellier, France*
[3]*Seismology and Geophysics Division, Department of Geology and Mines, Post Box 173, 9*
*Thimphu, Bhutan*





**ABSTRACT**
In spite of an increasing number of paleoseismic studies carried out over the last decade along
the Himalayan arc, the chronology of historical and pre-historical earthquakes is still poorly
constrained. In this paper, we present geomorphologic and paleoseismic studies conducted over
a large river-cut exposure along the Main Fontal Thrust in southwestern Bhutan. The Piping
site reveals a 30-m-high fault-propagation fold deforming late Holocene alluvial deposits.
There, we carried out detailed paleoseismic investigations and built a chronological framework
on the basis of 22 detrital charcoal samples submitted to radiocarbon dating. Our analysis
reveals the occurrence of at least five large and great earthquakes between 485 ± 125 BC and
AD 1714 with an average recurrence interval of 550 ± 211 yr. Co-seismic slip values for most
events reach at least 13 m and suggest associated magnitudes are in the range of Mw 8.5-9. The
cumulative deformation yields an average slip rate of 25.3 ± 4 mm/yr along the Main Frontal
Thrust, over the last 2600 yr in agreement with geodetic and geomorphological results obtained
nearby.





## 1. INTRODUCTION

The Himalayas, accommodating ~50% of the India-Eurasia collision at a shortening rate of ~20 mm/yr [e.g. Lavé and Avouac, 2000; Ader et al., 2012; Burgess et al., 2012; Marechal et al., 2016], are a region of sustained seismicity as illustrated recently by the 2015 Mw 7.8 Gorkha earthquake in Nepal [e.g. Avouac et al., 2015; Grandin et al., 2015]. Instrumental and historical records indicate that similar and significantly larger earthquakes have occurred along the Himalayan arc since medieval times [e.g. Rajendran and Rajendran, 2005; Sapkota et al., 2013; Yule et al., 2006; Kumar et al., 2010; Bollinger et al., 2014; Hetenyi et al., 2016]. Records of earlier events are documented as well from man-made and natural paleoseismic exposures (Fig. 1a) [e.g. Nakata et al., 1998; Upreti et al., 2000; Lavé et al., 2005; Yule et al., 2006; Kumar et al., 2010; Mugnier et al., 2013; Sapkota et al., 2013; Bollinger et al., 2014; Berthet et al., 2014; Rajendran et al., 2015; Mishra et al., 2016; Le Roux-Mallouf et al., 2016; Wesnousky et al., 2017 ; Wesnousky et al., 2019].

A robust estimate of size and recurrence interval needs to extend the time period covered by this catalog of historical events over numerous seismic cycles. With the exception of the study by Bollinger et al. [2014] that yielded five events (and two inferred) from a discontinuous stratigraphic record assembled from four sites, other exposures have only revealed one to two events per site, and a total of a dozen distinct events for the ~2500-km-long Himalayan Arc. Even the Bollinger et al.'s study constitutes a rather short catalog when compared to data available for smaller structures such as the ~1300-km-long San Andreas Fault or the ~1000-km-long Dead Sea Fault or North-Anatolian Fault [e.g. Meghraoui et al., 2012; Rockwell et al. 2015]. This issue is mostly due to the accommodation of a high shortening rate along the frontal thrust faults leading to surface ruptures with vertical offsets of up to 10 m [e.g. Kumar et al., 2010; Le Roux-Mallouf et al., 2016] and an average recurrence interval of 500-1000 years [e.g. Bollinger et al., 2014]. Hence, to retrieve long event series, excavations need to reach extraordinarily large dimensions into young unconsolidated deposits, which poses arduous logistics and safety challenges.

In this study, in order to investigate large Himalayan earthquake series, we selected a site in southwestern Bhutan where a ~30-m-high natural section is exposed by erosion at the outlet of a trans-Himalayan river called the Wang Chu. After describing the Bhutan Himalaya setting, we present the geomorphological and paleoseismic investigations carried out around and along this exposure. Our results allow us to discuss the timing and the magnitude of five surface-rupturing events that occurred in Bhutan during the last 2600 years.



## 2. MORPHOTECTONIC SETTING

### 2.1. Active tectonics in Bhutan

From north to south, Bhutan can be divided into four distinct tectonic units (Fig. 1b): the Tethyan Sedimentary Series (TSS), the Higher Himalaya (HH), the lesser Himalaya (LH), and the Siwaliks (Sw). All these units are bounded by major faults including the South Tibetan Detachment (STD), the Main Central Thrust (MCT), the Main Boundary Thrust (MBT), and the Main Frontal Thrust (MFT), which is the most recent expression of the thrust sequence that accommodated the deformation over geological time scales [Gansser, 1964; Le Fort, 1975; McQuarrie et al., 2008; Long et al., 2011a]. At depth, these four major north-dipping thrust faults connect to the Main Himalayan Thrust (MHT), a mid-crustal decollement under which the Indian plate subducts beneath the Himalayas and Tibet. In terms of geometry, several studies suggest a ramp-flat-ramp geometry of the MHT [e.g., Zhao et al., 1993; Nelson et al., 1996; Cattin and Avouac, 2000; Nábelek et al., 2009, Coutand et al., 2014, Le Roux-Mallouf et al., 2015].

Present-day deformation is constrained by (1) a far-field convergence of $17 \pm 0.5$ mm/yr inferred from geodetic measurements along 3 profiles across western, central and eastern Bhutan [Marechal et al., 2016] and (2) a single estimate of Holocene uplift rate of $8.8 \pm 2.1$ mm/yr, from the study of alluvial terraces along the front in central Bhutan [Berthet et al., 2014]. A first paleoseismic study by Le Roux-Mallouf et al. [2016] suggests that south-central Bhutan has been struck by at least two earthquakes during the last millennium, including (1) a Mw 7.5-8.5 earthquake in central Bhutan that produced ~1 m of coseismic uplift in AD 1714 [see also Hetényi et al., 2016] and (2) a Mw > 8.5 earthquake that produced ~8 m of coseismic uplift during the medieval times (between AD 1204 and AD 1464). This last event contributes to the debate about the possible deficit of seismic moment along the Himalayan arc [e.g. Bilham et al., 2001; Stevens and Avouac, 2016] and the probability of occurrence of a subduction-type Mw 9 earthquake in this region [Kumar et al., 2010; Mugnier et al., 2013; Srivastava et al., 2013; Stevens and Avouac, 2016, Le Roux-Mallouf et al., 2016, Wesnousky et al., 2017].

### 2.2 Geomorphology of the study area

The study site, called Piping, is located in the Lhamoizingkha area (SW Bhutan) immediately upstream of the confluence between the Wang Chu and the Ramphu Chu, a 5-km-long tributary that drains a 4.5-km² watershed (Fig. 2a). There, the MFT crosses the Wang Chu (89.759980°E, 26.722853°N) and a river-cut exposure reveals geological units and structures (Fig. 2b & 3):





93  - *The Lesser Himalayan zone-LH* (Manas Formation, Neoproterozoic-Cambrian) in the

94   north, composed of quartzite, phyllite and dolostone [Long et al., 2011a and references

95   therein] dipping 70-80° to the north;

96  - *The Subhimalayan zone-S* (Siwaliks, Miocene-Pliocene), immediately north of the

97   MFT, composed of medium-to-coarse-grained sandstone and pebble-to-cobble-

98   conglomeratic sandstone [Long et al., 2011b et references therein] dipping 50-70° to

99   the north and visible over more than 300 m;

100  - *The Alluvial plain,* composed of young unconsolidated sediment.

The MFT separates the flat, mostly undeformed, deposits of the Alluvial plain to the south from
a well-developed 4-km-long flight of alluvial terraces deposited by the Wang Chu over the
Manas and Siwalik formations. These terraces are composed of well-stratified cobbles to
boulders (dominant lithology is metamorphic from the Manas Formation) with a sandy matrix.
Available outcrops display relatively thin sediment covers (generally less than 6 m) deposited
over clear strath surfaces cutting into the Manas and Siwaliks Formations. The lower terraces
(T1, T2 and T3) are located directly along the present stream at low relative elevations (~1 m,
~11 m and ~33 m, respectively). T1 and T2 are deposited over the fault trace (Fig. 2a) and
display continuous top surfaces suggesting no significant deformation occurred since their
deposition. T1 is likely immerged during the monsoon season, as attested by natural and
anthropic detritus caught in the low vegetation. Intermediate terraces (T4, T5 and T6) appear
as continuous ribbons perched above the present river level at ~43 m, ~80m and ~90 m,
respectively. Finally, higher terraces T8 and T9 are strongly dissected and preserved as thick
alluvial sequences (e.g. ~18-m-thick for T8) on top of steep buttes forming local heights at
~100 m and ~170 m above the present river level, respectively.
East of the study site, a local watershed basin called Ramphu Chu cuts into the Manas and
Siwaliks formations and exits the steep piedmont at the location of the MFT where it forms a
500-m-wide alluvial fan (Fig. 2a). The upstream section of the fan was deposited against the
main MFT tectonic scarp and over the fault trace as visible on field photographs (Fig. 2b and
3a) and provides the main stratigraphic section studied here to unravel the recent deformation
history along the MFT.

**3. PALEOSEISMIC EXPOSURE**
An orthorectified photographic mosaic (Fig. 3a) of the site shows the 30-m-high river-cut cliff
and displays a 40-m-wide deformation zone that separates the grey Siwaliks (unit S) to the
north, topped by the south-dipping U7 terrace (Wang Chu deposits) from an horizontal 18-m-





thick sequence of fan deposits (U6 to U0) from the Ramphu Chu. A 50-m-long by 30-m-high
section of the natural exposure was cleaned, partly gridded and logged in details (Fig. 3b and
following) based on stratigraphy, lithology and grain-size. Overall, 50 samples of organic
matter (charcoal and plant debris) were collected, and 22 were selected for radiocarbon age
determination (Table 1).

**3.1. Chronostratigraphy**
The stratigraphy of the northern section of the exposure (Fig. 3) is mostly constituted of
massive grey sands with fine beds of white silts, pebbles and cobbles that outline a ~60° dip to
the north. This unit crops out along a ~150-m-long section of the river cut and exhibits a
thickness of at least 90 m. It is widely observed regionally along the mountain front (Long et
al., 2011a) and is attributed to the Siwaliks formation (S). Here, it is overlain with a ~4-m-thick
clast-supported stratified cobbles-to-boulders unit (called U7 hereafter). Observed clasts are
generally rounded with a significant contribution of metamorphic lithology from the Higher
Himalaya formation (Long et al., 2011a). Considering stratigraphy, clasts roundness, distance
to the nearest outcrops of said formation (~25 km north of the site) and relationship to the local
drainages, we interpret this unit as an alluvial terrace deposit from the trans-Himalayan Wang
Chu. Unit U7 is stratigraphically above the Siwaliks (S) and lies over a clear erosion surface
(strath) that cuts through the Siwaliks north-dipping stratigraphy. Its top surface is eroded north
of grid point (22, 24) and preserved and overlain with a succession of fine-grained units south
of it (Fig. 3b); it is hereafter considered to mark the base of the Quaternary stratigraphic record
at this site.
On top of unit U7, we observed an 18-m-thick succession of deposits comprised of 20-to-40-
cm-thick massive bluish-grey silt layers and clast-supported gravel layers with a sandy matrix.
Major sediment packages are delimited along continuous near-horizontal (in the undeformed
section) limits and named U6 (deepest) to U0 (shallowest). They exhibit abundant detrital
charcoal lumps, most of them reaching 1 cm in diameter and displaying freshness, compactness
and angularity indicative of *a priori* short transport and storage times. Overall, 50 samples were
collected from units U6 to U0, of which 22 were selected and submitted for radiocarbon dating
(Table 1). Fine calibration was performed with OxCal 4.2 using a depositional model where
samples from the same unit are defined as a phase [*e.g.* Lienkaemper & Bronk Ramsey, 2009]
and yielded dates consistent with the observed stratigraphic order.

- Unit U6: the lowest unit lies over unit U7 over the northern section of the exposure

(north of x=22) where it is ~2 m thick, while its base is presently below the water table in the



southern section and could not be logged (Fig. 3). It is comprised of massive fine to very fine
silts, blueish grey in color, interbedded with 30-to-40-cm-thick poorly stratified lenses of
matrix-supported angular gravels, containing ~50% of fine to coarse sand. The top of U6 is
marked by a relatively smooth poorly expressed erosion surface. The age of the unit is
constrained by 7 samples with a narrow distribution of radiocarbon ages comprised between
$2480 \pm 30$ yr BP and $2625 \pm 30$ yr BP (Table 1) suggesting a relatively fast deposition process.
A single obvious outlier (sample PI-C46 with a radiocarbon age of $37700 \pm 800$ yr BP) was
considered reworked, and therefore discarded from our analysis. Model calibration yields a
deposition date of $670 \pm 165$ BC.

- Unit U5: within the southern undeformed section of the exposure section, this unit

displays a thickness of ~1.5 m (south of x=59 m in Fig. 3b). It exhibits a similar grain-size
distribution to that of U6 but with distinct gravel and sand lenses: the bottom section is marked
by well-defined fine gravel lenses while the top section is evidenced by a ~1-m-thick coarse
sand and gravel lens. The top of unit U5 is defined by a weakly-expressed erosional surface
that probably reflects more a short depositional hiatus rather than established rill processes.
Unit U5 yielded 6 samples, 4 of which with ages between $2180 \pm 30$ yr BP and $2285 \pm 30$ yr
BP, again indicative of a relatively fast deposition process. The two remaining samples
collected at the base of the unit (PI-C11 and PI-C12) are significantly older than other samples
from U5 and even U6 ($2905 \pm 30$ yr BP and $2860 \pm 30$ yr BP, respectively). We suspect they
have been reworked from the lower section of U6 or from an even older unit, and we choose
therefore to discard them from our analysis. Model calibration yields a deposition date of 290
$\pm 120$ BC.

- Unit U4: this unit is 3 to 4 m thick in the southern section of the exposure (south of

x=55 m in Fig. 3b) and thins out to the north where is forms an onlap against U5 then U6 at
x=38 m. U4 is almost entirely composed of matrix-supported gravels with a few silt lenses and
terminates with a continuous ~15-cm-thick sand layer. This unit did not yield any adequate
sample for radiocarbon dating, probably on the account of the higher energy regime at the time
of its formation.

- Unit U3: this unit displays a very constant thickness of ~1.5 m over the whole exposure

(between x=24 and x=98). It is comprised of massive silts with 20-to-30-cm-thick lenses of
coarse sand and fine gravel. U3 yielded 3 samples with radiocarbon ages of $1730 \pm 30$ yr BP,
$1960 \pm 30$ yr BP and $2560 \pm 30$ yr BP. Since the latter sample is contemporaneous of U6, it is
considered reworked and removed from any subsequent analysis. Model calibration yields a
deposition date of AD $240 \pm 100$.





- Unit U2: this unit also exhibits a constant thickness of ~1.5 m over the exposure. It is,
however, comprised of matrix-supported gravels with a few sand lenses, which suggests a
slightly higher energy fluvial regime. It yielded 3 samples with radiocarbon ages 1520 ± 30 yr
BP, 1770 ± 30 yr BP and 2405 ± 30 yr BP. Similarly, since the latter is contemporaneous of
U6, it is considered reworked and removed from subsequent analysis. Model calibration yields
a deposition date of AD 440 ± 70.
- Unit U1: this unit is ~3 m thick over the exposure. It displays a stratigraphic content
very similar to that of unit U2 and lies over a weak erosional surface forming the top of U2.
For logistics and safety reasons, unit U1 could not be sampled for age determination.
- Unit U0: this is the ultimate deposit of this section. It displays a variable thickness of
~20 cm to up to 4.5 m with a strongly eroded top surface within the deformed zone, north of
x=52 m (Fig. 3b). The top of U0 marks the abandonment of the section before it was intensely
and almost entirely incised by a local gully (x=52-70 m). Although this unit was directly
accessed at the location of the uppermost log (box marked "Fig. 8" in Fig. 3), we could not
retrieved adequate material for age determination.

Within this succession, clasts lithology and roundness are constant, thus suggesting a common
nearby source for units U6 to U0 distinct from that of U7. Gravels are very angular and made
of quartzite and phyllite from the Manas Formation, sands are fine-grained and well classed
and silts are massive and blueish gray in color, where not oxidized. Although grain size
distribution varies across units from gravel-dominant (with sand lenses) to silt-dominant (with
sand and gravel lenses), this does not necessarily reflect significant variations in transport flow
velocity [e.g. Miller et al., 2014]. Overall, we interpret units U6 to U0 to derive from the same
nearby low-flow-velocity source consistent with the recent alluvial fan mapped at the outlet of
the Rampu Chu watershed basin (Fig. 2).

Two additional units display specific wedge-shaped geometries: W2 between U5 and U4 and
W1 deposited against U0 and immediately below the modern soil. Both units exhibit little
stratigraphy, intense internal deformation (see details below) and are interpreted as colluvial
wedges (more details in the following section). W1 is stratigraphically the youngest unit
observed here. Two detrital wood samples (PI-C23 and PI-C24) yield modern ages. Since roots
found in the region sometimes resemble tree-trunk bark in terms of size, density and texture,
we suspect the ligneous samples PI-C23 and PI-C24 may derive from in-situ roots and may not
be representative of W1's true age. These samples are discarded in our analysis.




229 Additionally, it is quite notable that the undeformed part of the 18-m-thick Ramphu Chu section

230 (south of x = 54 in Figure 3b) presents a quasi-continuous (erosion surfaces are poorly

231 expressed and stratigraphic limits are virtually flat) succession of silt, sand and gravel deposits

232 constrained by 15 radiocarbon samples (Table 1). To better assess the timing of deposition for

233 the uppermost units, we assume that deposition was mostly continuous and we build an age-

234 versus-height relationship for all samples retained for our analysis (Figure 4). Our approach

235 yields an average deposition rate of 7.1 ± 0.2 mm/yr between 805 ± 30 BC (U6) and AD 520

236 ± 95 (U2), with potential short-term variability between silt and gravel beds [e.g. Kumar et al.

237 2007]. On that basis, and considering a similar constant sedimentation rate until the final

238 deposition of U0, we may extrapolate the deposition rate and propose a tentative date with large

239 uncertainties (2σ) for the top of U1 at AD 940 ± 200. Since U0 is strongly eroded, we did not

240 attempt to date its top surface.

## 3.2. Exposure description

243 Large-scale deformation across the MFT at the Piping site is illustrated by fault-propagation

244 folding affecting terrace unit U7 shown on Figure 3. U7 crops out ~34 m above the present

245 stream (grid point (0, 34) in Fig. 3b), dips increasingly to the south, is sheared by a system of

246 north-dipping thrust fault splays (F2 to F5 in Fig. 3b), dips reverse to the north and disappears

247 underneath a massive 8-to-10-m-thick fault gouge (unit G in Fig. 3b and following). Since U7

248 does not crop out south of the main fault zone, it is necessarily deeper than the present river

249 level (at least below U6) and has hence recorded more than 34 m of uplift since its deposition.

250 Subsequent units U6 to U0 are mostly undeformed from the southernmost tip of the exposure

251 to the center of the studied section (i.e. south of x = 54 m in Fig. 3b). There, they exhibit various

252 stages of deformation, from warping with minor faulting (U0 to U3) to folding (U4) and intense

253 faulting with duplexing (U5 and U6), indicating than the older units of the Ramphu Chu fan

254 have cumulated more deformation. Furthermore, fault strand F5 cuts through the whole section

255 and reaches the surface with a near-vertical dip and affects U2 to U0 with an apparent normal

256 geometry. To describe faulting and abutting relationships in detail and identify surface-

257 rupturing events, we focus on two excerpts presented at high resolution in Figures 5 to 8.

258 The lower section documents deformation affecting units U7, U6 and U5 (Fig. 5, 6 and 7).

259 From grid point (28, 2) (Fig. 5b), U7 is overlain with unit (G) composed of massive reddish to

260 brownish clay that contains sheared and fractured clasts from the Siwaliks formation as well as

261 cobbles and boulders from U7. It exhibits intense internal deformation (see close-up in Fig. 7a)

262 typical of a fault gouge. The localized fault contact between G and U7 corresponds to F4 in





Fig. 3b and Fig. 5b. To the south, U6 crops out at the base of the exposure and is affected by
fault F1, which cuts through U6 and U5, and dies out ~4 m southward within U5 (Fig. 5b). F1
accommodates only minor faulting as attested by a relatively small 30-cm offset affecting the
base of U5 (Fig. 7b). Secondary normal-geometry splays F6 and F7 branch out from F1 and
displace the base of U5 vertically by a total of ~60 cm. F7 tapers out within U5 while F6 cuts
it entirely and terminates against the low-dipping fault strand F2 at a right angle. Above F2,
U6 displays strongly-deformed near-vertical bedding produced by dragging along F2 (Fig. 6)
and forms a fault-propagation fold. Hence, F2 is a duplex fault that accommodates major
deformation within the exposure. The uppermost part of unit U6 is affected by similar
duplexing deformation along the F3 fault strand, though with a much smaller offset. F2 also
affects U5 where duplexing produced a clear scarp overlain with wedge-shaped unit W2. Its
stratigraphy is composed of finely-layered silts and gravels similar to U5 but exhibits intense
deformation with sheath folds typically associated with slumping along a slope (Fig. 7c), here
consistent with the frontal slope of the scarp. We interpret W2 as a scarp-derived colluvial
wedge deposited during or shortly after a co-seismic displacement along F2 affecting U5. The
top of W2 and U5 are in continuation and overlain by U4, which does not exhibit noticeable
deformation at this location and show that F2 was not re-activated after the deposition of U4.
The upper section (Fig. 8) documents the northernmost fault strands F4 and F5 as they reach
the surface. At the bottom of the trench (Fig. 3b), F4 and F5 originate from the main gouge
zone (G) where they dip ~20°N, cut through U7 with a steeper dip of ~50°N and merge together
as strand F4/F5, cut through U3 at a near-vertical angle and U2 to U0 with a ~85°S dip. This
change of dip angle and direction is expressed within the shallowest units (U3 to U0) by an
apparent normal-geometry fault displacement along F4/F5 (see Fig.8b). The detailed log of the
upper section shows a ~3-m-wide V-shaped deformation zone bounded by F4/F5 to the north
and by a diffuse deformation band affecting U3 to U0 to the south (x = 38-41 m in Figures 3b
and 8). In between, units exhibit strong warping and chaotic limits suggesting soft-sediment
deformation and collapse against F4/F5. Unit U1 is overlain with U0, which is itself collapsed
against F4/F5. The amount of associated vertical displacement is difficult to ascertain, due to
the wide collapse zone and the fact that U0 has been eroded north of F4/F5. From the base of
the hanging wall section of U1 at grid point (37, 18) to the base of the footwall section of U1
at grid point (38.5, 16.5), we estimate a minimum vertical offset of ~1.5 m. Finally, the whole
stratigraphic succession is sealed by a ~1.5-m-thick wedge-shaped colluvial unit (W1)
deposited over U0 and against what we interpret as F4/F5 free face.





### 3.3. Timing of surface ruptures and associated co-seismic displacements

In order to identify the various deposition, erosion and deformation events recorded at the Piping site, we propose a schematic sequential retro-deformation combining all observations collected over the exposure (Fig. 9; see Malik et al., 2017, for a similar approach further west). We start from a simplified log (Fig. 9a) and successively retro-deform the whole section to restore the most recent deposits to their original geometry and infer previous events where deformation remains. In parallel, we present OxCal-modeled [Bronk Ramsey, 2009] event dates constrained by 15 radiocarbon samples (see section 3.1) and a chronostratigraphic model following guidelines from Lienkaemper and Bronk Ramsey [2009] (Fig. 10):

- Event 1: The most recent deposit observed in the exposure is a ~1.5-m-thick colluvial wedge (W1 in Fig. 8 and 9a) deposited against a free face formed in unit U1 by slip along F4/F5. The diffuse deformation observed within U3, U2 and U1 and the collapse of unit U0 within an open fissure are contemporaneous with a first event that occurred after the deposition of U0 (Fig. 8). Radiocarbon-dating of W1 only yielded modern dates (Table 1) -likely due to contamination from actively developing soil- and does not permit to date E1 accurately. From our chronostratigraphic analysis (Fig. 10), E1 occurred after AD 895 and was associated with faulting along faults F4 and F5. Removal of W1 and retro-deformation of units U0 to U3 restore the continuity of the bottom of U0 and leave large-scale folding affecting units U2 and older (given the poor constraint on the size of this first event, which seems a priori small with a minimum displacement of ~1.5 m, we did not represent the stage before event E1).

- Event 2: Large-scale folding deforms units U2 to U0 uniformly (Fig. 9a) and indicates a major deformation event affected the stratigraphic section after the deposition of units U2 to U0. Restoring these deposits to their original horizontal geometry (Fig. 9b) in agreement with the southern section of the exposure (Fig. 3b) involves (at least) bringing the highest observable point of unit U1 (erosion surface at grid point (26, 25.5) marked by the northern green star in Fig. 9b) down to the height of U1 top observed in the undeformed section (e.g. grid point (50, 14) marked by the southern green star in Fig. 9b). This analysis yields a minimum cumulative (E1+E2) vertical offset of ~11.5 m along the 50-90° north-dipping F4-F5 splay. Considering an average dip of 60° and a co-seismic slip of ~1.5 m for E1, the net co-seismic dip-slip for E2 reaches at least 12 m. Furthermore, our chronostratigraphic model (Fig. 10) yields the same time window for the occurrence of E2 as for E1, i.e. after AD 895. Removing the now undeformed units U2 to U0 reveals that significant folding and faulting remain for units U3 and older (Fig. 9c).





- Event 3: By applying the same approach to units U4 and U3 and considering that the uppermost point of the top of unit U3 has been eroded away, we estimate the height difference between grid point (26, 25.5) and the height of the top of U3 in the undeformed section, to be 9.5 m (blue stars in Fig. 9c). This yields a minimum cumulative vertical offset along F3, F4 and F5 of ~16 m for E3+E2+E1, hence ~4.5 m of vertical offset for E3 alone. Since slip propagated primarily along F3 with an average dip of ~20°, we estimate the co-seismic dip-slip for E3 along F4 at ~13.2 m. U3 is the youngest affected unit, while U2 is the oldest unaffected unit, which indicates E3 occurred between the deposition of U3 and U2. Our radiocarbon chronology (Fig. 10) yields a date of occurrence at AD 300 ± 70. Retro-deformation along F3, then removal of undeformed units U4 and U3 suggests residual deformation affects units U5 and older (Fig. 9d).

- Event 4: At this stage (Fig. 9e), units U5 and U6 form a ~2-m-high scarp on the ground surface rapidly covered by scarp-derived colluvium W2 at the toe of the scarp. In Figures 3 and 5, the U5 package located underneath F2 between x=33.5 m and x=38 m only exhibits the lower part of U5 (units U5b and U5c) while the duplexed part above F2 only exhibits the upper section of U5 (U5a). Hence, restoring U5 involves removing W2 then retro-sliding the duplexed part of U5 along F2 to bring grid point (51.5, 5) back to its minimal original position at grid point (39.5, 4) with a dip-slip offset of ~13.5 m along F2. In parallel, minor displacements along F1 (~30 cm reverse faulting, see Fig. 7b), F6 (~25 cm normal faulting) and F7 (~35 cm normal faulting) accommodate the anticlockwise rotation of a ~10 m long block of U5 and U6 underneath F2, likely associated with pure shear deformation under the weight of the propagating fold (see Fig.5b). This event is predated by the deposition of U5 and postdated by the deposition of U4, hence bracketed at 100 ± 160 BC (Fig. 10). This brings U5 to its original undeformed geometry forming a near horizontal unit deposited against a pre-existing scarp formed in U6, as attested by the onlap termination visible at grid point (38, 13) in Figure 3a.

- Event 5: This event is documented by the remaining scarp affecting U6 once previous events are retro-deformed and U5 is removed (Fig. 9g). Although the height of this scarp is poorly constrained, the retro-deformation analysis suggests it is at least 2 m high and was produced by slip along a shallow-dipping rupture (~10°N), similar to F2 and F3 as observed at the base of the exposure (below z=1 m). Hence, we propose that the amount of slip involved during E5 is similar to what is inferred for E4. Furthermore, since the event took place between the deposition of units U6 and U5, it may be dated back to 485 ± 125 BC (Fig. 10).



A striking feature of surface deformation visible in the Piping exposure is the gradual change
in fault dip over time. While all fault strands converge and dip 35-40°N below grid point (30,
2) they diverge from ~10°N to ~50°N (locally 90°) as they propagate to the south (Fig. 3 and
Fig. 9),presenting a geometry similar to tri-shear folding [Allmendinger, 1998]. In detail, the
oldest event (E5) occurred while the top of unit U6 constituted the ground surface (i.e. the event
horizon) and is expressed along a shallow 10° north-dipping duplex rupture. The situation is
similar for E4. After deposition of units U4 and U3 adding 2.5-3 m of sediments on top of the
E5 rupture, the following event (E3) emerges higher in the stratigraphic section along F3 with
a steeper dip of 25-30°. A consequent deposition episode adds at least 8.5 m of sediments (units
U2, U1 and U0) over these ruptures. The most recent event(s) (E2 and E1) exhibit a much
steeper rupture (along strands F4 and F5) with a dip reaching ~50° within unit U7 (coarse-
grained terrace deposits) and 90° as it emerges to the present-day surface through unit U0 (fine-
grained fan deposits).
It is a common observation both in the field and in analog experiments that ruptures along
thrust faults tend to flatten as they reach the surface under the influence of decreasing lithostatic
pressure [e.g. Philip and Meghraoui, 1983; Lee et al., 2001]. We propose that the change in
deformation style from nearly horizontal (E5 and E4) to steep (E2) then vertical (E1) displayed
in the Piping trench reflects increasing vertical load onto the foot of the tectonic scarp
associated with the progressive buildup of the Rampu Chu fan against it.


**4. SUMMARY OF RECURRENCE TIMES, MAGNITUDES AND SLIP RATE**
Paleoseismic investigations conducted along the MFT at the confluence between the Wang
Chu and the Ramphu Chu in Western Bhutan show an important cumulative deformation zone
including a rich chronology of deposition phases and deformation events for the last ~2600
years.
The most recent event (E1) is consistent in terms of amount of co-seismic slip and
chronology with the most recent event identified by Berthet et al. [2014] and Le Roux-Mallouf
et al. [2016] in the Sarpang area (~50 km to the east, see Fig. 11) and interpreted as the AD
1714 earthquake (previously described as the AD 1713 earthquake by Ambraseys and Jackson,
2003). By combining historical and paleoseismic constraints, Hétényi et al. [2016] propose that
this earthquake reached Mw 7.5-8.5 with a modeled rupture centered on Bhutan and largely
encompassing the Piping site. We therefore propose that E1 corresponds to the AD 1714
earthquake. Similarly, our event E2 is consistent with an event observed at the Sarpang site as



well dated AD 1344 ± 130 (Fig. 11) and tentatively associated with a medieval earthquake that
may have ruptured a large section of the MFT (see Le Roux-Mallouf et al., 2016 and references
therein). Hence, we propose that our event E2 corresponds to that second event. Events E3, E4,
and E5 occurred at AD 300 ± 70, 100 ± 160 BC, and 485 ± 125 BC, respectively.
Hence, according to our retro-deformation analysis and chronostratigraphic model, our results
allow constraining the occurrence of five surface-rupturing events between 485 ± 125 BC and
AD 1714 with an average recurrence interval of 550 ± 211 yr. When only considering events
with the largest documented co-seismic slip values (E2 to E5) that are the most likely to be
preserved and observed in exposures, the average recurrence interval reaches 610 ± 238 yr.
Ours results are comparable to the lower values obtained for the late Holocene by Bollinger et
al. [2014] in eastern Nepal (610 to 1220 yr, depending on hypotheses). Furthermore, the
relatively small co-seismic slip value determined for E1 (and assigned to the AD 1714
earthquake) suggests smaller though destructive events may occur on occasion as was the case
for the 2015 Gorkha earthquake in Central Nepal [e.g. Grandin et al., 2015] although there was
no surface rupture associated with it.
The retro-deformation analysis also allows estimating associated dip-slip co-seismic
displacements with values ranging from ~1.5 m for E1 to more than 13 m for E2, E3, E4 and
probably E5, a value typical of the largest events documented along the Himalayas in Nepal,
Sikkim, Bhutan and Assam and consistent with extreme magnitudes on the order of $M_w$ 9 (Le
Roux-Mallouf et al., 2016 and references therein). Considering the largest events, this
represents ~40.2 m of slip (E2+E3+E4) accrued over 1629 ± 255 yr (between E5 and E2) at a
rate of 25.3 ± 4 mm/yr. Although the duration of our dataset may be too limited to represent
the long term behavior of the MFT, this slip rate is consistent with those derived from 8-kyr-
old uplifted terraces in Sarpang (Fig. 11) [Berthet et al., [2014] and from far-field GPS
shortening rate measurements [Marechal et al., 2016]. Together, these results suggest that the
Himalayan convergence is mainly seismically accommodated along the MFT in western
Bhutan as well.


**5. CONCLUSION**
We presented here the longest continuous record of paleo-earthquakes along the Himalayan
arc from the detailed study of an 18-m-thick deformed sedimentary sequence dated from 15
radiocarbon samples. Well-expressed deformation and a detailed retro-deformation analysis
reveal the occurrence of five surface-rupturing earthquakes along the MFT in southwestern



Bhutan during the past ~2600 years. The two most recent events can be related to the AD 1714
earthquake [Hétényi et al., 2016] and a medieval event (AD 1344 ± 130) already described in
south central Bhutan [Le Roux-Mallouf et al., 2016]. More strikingly, events E3, E4 and E5
are documented here for the first time and constitute some of the oldest paleoearthquakes
characterized in the Central Himalayas (Fig. 11). Together, these events give an average
earthquake recurrence interval of 550 ± 211 yr (or 610 ± 238 yr for the largest) for the Main
Frontal Thrust in Bhutan.
The slip rate of 25.3 ± 4 mm/yr obtained from cumulative slip is consistent with both Holocene
rates obtained from uplifted terraces [Berthet et al., 2014] and high interseismic coupling level
inferred from geodetic measurements [Marechal et al., 2016], which suggests that the
Himalayan convergence in western Bhutan is mainly seismically accommodated along the
MFT. Moreover, this result suggests that –at least locally- the slip budget does not display
significant deficit over the time period of this study [Stevens and Avouac, 2016]. Finally,
estimated co-seismic displacements between ~1.5 m and at least 13 m indicate the occurrence
of large (between Mw ~7.5 and Mw ~8.5) and great earthquakes (MW > 8.5) at a single site.
This complexity should be taken into account in probabilistic seismic hazard calculations.

**Author contribution**
RLM, MF, JFR and PP conducted field work. RLM and MF prepared the manuscript with
contributions from all co-authors.

**Competing interests**
The authors declare that they have no conflict of interest.

**ACKNOWLEDGMENTS**
This project is funded by the French Agence Nationale de la Recherche (ANR-13-BS06-0006-
01) and CNES (Pleiades satellite images and field support). We would like to thank all people
helping in the field and particularly our driver Phajo Kinley from the Department of Geology
and Mines. We also thank S. Dominguez (Géosciences Montpellier) for fruitful discussions.
Correspondence and requests for materials should be addressed to
romain.lerouxmallouf@geolithe.com





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

Preliminary stratigraphic and structural architecture of Bhutan: Implications for the along
strike architecture of the Himalayan system: *Earth and Planetary Science Letters*, 272(1),
105-117, doi: 10.1016/j.epsl.2008.04.030.
Meghraoui, M., Aksoy, M. E., Akyüz, H. S., Ferry, M., Dikbaş, A., & Altunel, E. (2012).
Paleoseismology of the North Anatolian fault at Güzelköy (Ganos segment, Turkey): Size
and recurrence time of earthquake ruptures west of the Sea of Marmara. Geochemistry,
Geophysics, Geosystems, 13(4).
Miller, K.L., Reitz, M.D. & Jerolmack, D.J. (2014). Generalized sorting profile of alluvial fans.
*Geophysical Research Letters,* 41, 7191-7199. doi: 10.1002/2014gl060991.
Mishra, R. L., Singh, I., Pandey, A., Rao, P. S., Sahoo, H. K., & Jayangondaperumal, R. (2016).
Paleoseismic evidence of a giant medieval earthquake in the eastern Himalaya. *Geophysical*
*Research Letters*, *43*(11), 5707-5715.
Mugnier, J.-L., Gajurel, A., Huyghe, P., Jayangandaperumal, R., Jouanne, F., Upreti, B.,
(2013), Structural interpretation of the great earthquakes of the last millennium in the central
Himalaya, *Earth Science Reviews*, *127*, 30-47, doi: 10.1016/j.earscirev.2013.09.003.





Nakata, T., Kumura, K., & Rockwell, T. (1998). First successful paleoseismic trench study on
active faults in the Himalaya. *Eos Trans. AGU*, *79*, 45.
Nelson, K. D., Zhao, W., Brown, L. D., & Kuo, J. (1996). Partially molten middle crust beneath
southern Tibet: synthesis of project INDEPTH results. *Science*, *274*(5293), 1684.
Philip H., and Meghraoui, M. (1983). Structural analysis and interpretation of the surface
deformations of the El Asnam Earthquake of October 10, 1980. *Tectonics*, 2, 17-49. doi:
10.1029/TC002i001p00017.
Rajendran, C. P., and K. Rajendran (2005), The status of central seismic gap: a perspective
based on the spatial and temporal aspects of the large Himalayan earthquakes,
*Tectonophysics*, *395*(1), 19–39, doi:10.1016/j.tecto.2004.09.009.
Rajendran, C. P., B. John, and K. Rajendran (2015), Medieval pulse of great earthquakes in the
central Himalaya: Viewing past activities on the frontal thrust, *J. Geophys. Res. Solid Earth*,
120(3), 1623–1641 doi:10.1002/2014JB011015.
Rockwell, T. K., Dawson, T. E., Ben-Horin, J. Y., & Seitz, G. (2015). A 21-event, 4,000-year
history of surface ruptures in the Anza seismic gap, San Jacinto Fault, and implications for
long-term earthquake production on a major plate boundary fault. *Pure and Applied*
*Geophysics*, *172*(5), 1143-1165.
Sapkota, S. N., Bollinger, L., Klinger, Y., Tapponnier, P., Gaudemer, Y., & Tiwari, D., (2013).
Primary surface ruptures of the great Himalayan earthquakes in 1934 and 1255. Nature
Geoscience, 6(2), 71-76, doi:10.1038/ngeo1669.
Srivastava, H. N., B. K. Bansal, and Mithila Verma (2013), Largest earthquake in Himalaya:
An appraisal, Journal of the Geological Society of India, 82.1, 15-22.
Stevens, V. L., and J.-P. Avouac (2016), Millenary Mw > 9.0 earthquakes required by geodetic
strain in the Himalaya, Geophys. Res. Lett., 43, 1118 – 1123, doi:10.1002/2015GL067336.
Upreti, B. N., Nakata, T., Kumahara, Y., Yagi, H., Okumura, K., Rockwell, T. K., Virdi, N.S.,
& Maemoku, H. (2000). The latest active faulting in southeast Nepal. *Active Fault Research*
*for the New Millennium*, 533-536.
Yule, D., J. Lave, S. N. Sapkota, D. Tiwari, B. Kafle, M. R. Pandey, S. Dawson, C. Madden,
and M. Attal (2006), Large surface rupture of the Main Frontal Thrust in east-central and
western Nepal-Evidence for an unprecedented type of Himalayan earthquake, Proceedings
on the International Workshop on Seismology, seismotectonics and seismic hazard in the
Himalayan region, Kathmandu, 28–29 November, 2006, 13–14.534.





Wesnousky, S. G., Kumahara, Y., Chamlagain, D., Pierce, I. K., Karki, A., & Gautam, D.

(2017). Geological observations on large earthquakes along the Himalayan frontal fault near

Kathmandu, Nepal. *Earth and Planetary Science Letters*, *457*, 366-375.

Wesnousky, S. G., Kumahara, Y., Chamlagain, D., & Neupane, P. C. (2019). Large Himalayan

Frontal Thrust paleoearthquake at Khayarmara in Eastern Nepal. *Journal of Asian Earth*

*Sciences,* 174, 346-351.








**TABLE**

**Table 1.** AMS Radiocarbon (14C) dates from detrital charcoals collected from the Piping

exposure. Samples in italics were discarded from our analysis (see main text for details).

[a]See trench log for stratigraphic unit designations.
[b]Radiocarbon years B.P. relative to 1950 AD (with 1 σ counting error). All samples have been dated by the
Poznan Radiocarbon Laboratory.
[c]Calendric dates were calibrated using OxCal and the atmospheric calibration curve IntCal13. Calendric ages
have been rounded to the nearest ½ decade assuming the 5 years accuracy of the IntCal13 curve. D Calendric
dates were calibrated using the atmospheric calibration curve IntCal09 for the Northern Hemisphere

**FIGURE CAPTIONS**
**Figure 1**
Location of the study area and its regional context. Inset shows the location of Bhutan along
the Himalayan arc. (A) Himalayan arc. Red stars are epicenters of great and large earthquakes
from instrumental, historical and paleoseismic studies. Orange rectangles are previous
paleoseismic studies (a) Mohana Khola [Yule et al., 2006]; (b) Koilabas Khola [Mugnier et al.,
2011]; (c & d) Tribeni and Bagmati [Wesnousky et al., 2017];(e) Sir Bardibas [Sapkota et al.,
2013; Bollinger et al., 2014]; (f) Khayarmara [Wesnousky et al., 2019]; (g) Marha Khola [Lavé
et al., 2005]; (h) Hokse [Nakata et al., 1998, Upreti et al., 2000]; (i) Panijhora [Mishra et al.,
2016] (j) Chalsa [Kumar et al., 2010]; (k) Sarpang [Le Roux-Mallouf et al., 2016]; (l) Nameri
[Kumar et al., 2010]; (m) Harmutty [Kumar et al., 2010]. The blue rectangle is the location of
the paleoseismic study presented in this paper. (B) North-south simplified geological cross
section across western Bhutan (modified after Grujic et al., [2011]). See Figure 1A for location,
dashed line labeled "b". Abbreviations are as follows: TSS, Tethyan Sedimentary Sequence;
HH, Higher Himalayan; LH, Lesser Himalayan; Sw, Siwaliks sediments; GP, Ganga Plain;
STD, Inner South Tibetan Detachment; KT, Kakhtang Thrust; MCT, Main Central Thrust;
MBT, Main Boundary Thrust; MFT, Main Frontal Thrust.

**Figure 2**
Geomorphological map of the study area. (A) Geomorphological map of the Main Frontal
Thrust, in the Piping area, superimposed on 2-m-resolution Pleiades-derived Digital Elevation





Model. Alluvial terraces are labeled from T0 (active channel) to T8 (oldest). Camera pictogram
indicates the location of the panorama in B. White star indicates the location of the Piping
exposure. Spacing of elevation contours is 20 m. Black dots indicate spot elevations extracted
from an in-house Pleiades DEM. (B) Panorama photography (eastward view) of the large scale
Piping site including the southern Piping exposure.

**Figure 3**
Piping paleoseismic exposure. (A) Orthorectified photomosaic of the left-bank of the Wang
Chu (southernmost section of Fig. 2b) showing the contact between the Siwalik units (light
grey) and Rampu Chu fan deposits (well-stratified beige to grey units). White rectangles
indicate the locations of Fig. 5, 7 and 8. (B) Detailed log over a 2-m grid. Solid and dashed red
lines are main faults (certain and suspected, respectively). Blue squares indicate the locations
and 2σ-calibrated calendar ages of 22 detrital charcoal samples. Samples in italics were
discarded from our analysis (see main text for details). The lower 1.5 m of the exposure is here
hidden by the access path built by the backhoe.

**Figure 4**
Evolution of age versus height for the Ramphu Chu sedimentary sequence. Data (black outline
diamonds) describes a satisfactory linear regression ($R^2 = 0.95$) and allows interpolating
towards present. Modelled points (red outline diamonds) and 2 σ variance determined from the
height of sedimentary limits suggest the top of U1 was deposited at AD 940 ± 200. Associated
uncertainties are deduced from the 2-σ curves.

**Figure 5**
Lower part of Piping paleoseismic exposure. (A) Orthorectified photomosaic of the left-bank
of the Wang Chu. White rectangles indicate the location of figures 6 and 7 (a, b and c). (B)
Detailed log over a 1-m grid. Solid and dashed red lines are main faults (certain and suspected,
respectively). Blue squares indicate the locations and 2σ-calibrated calendar ages of 22 detrital
charcoal samples. Samples in italics were discarded from our analysis (see main text for
details).





**Figure 6**

Enlarged photograph of the lower part of the exposure (see Fig. 5a for location) showing (1) sub-horizontal deposits of U5 and U6 below the thrust fault F2 and (2) the overturned limb of U6 and U7 characterized by tilted gravel and silty layers and pebbles, respectively.

**Figure 7**

Enlarged ortho-photographies showing (A) the northward-dipping-contact between the gouge fault G and the overturned alluvial terrace U7 at the bottom of the exposure, (B) the 50-cm-offset and the shear texture induced by the fold termination of the F1 thrust fault at the southern end of the deformation zone and (C) a slump figure within the colluvial wedge W2 associated with event E4 along fault splays F1 and F2.

**Figure 8**

Upper part of the Piping paleoseismic exposure. (A) Orthorectified photomosaic of the left-bank of the Wang Chu showing fault F4/F5 (B) Detailed log over a 1-m grid. Solid and dashed red lines are main faults (certain and suspected, respectively). F4/F5 is associated with a vertical fabric, affects all alluvial units and is capped by colluvial wedge W1. Blue squares indicate the locations and calendar ages of 2 detrital charcoal samples.

**Figure 9**

Sequentially restored cross section illustrating the chronology of the successive deposition and deformation episodes at the Piping site. All ages are derived from an OxCal chronostratigraphic model.

**Figure 10**

Chrono-stratigraphic model for deposition episodes (alluvial units U0 to U6 and colluvial wedge W1) and surface-rupturing events (E5 to E1) at the Piping exposure. The model is built from abutting relationships between stratigraphy and faulting and is constrained by 18 detrital charcoal samples and one inferred age corresponding to the top of unit U1. All resulting calendar dates are rounded to the nearest multiple of 5.

**Figure 11**

(A) Synthesis of available paleoseismic records along the Himalayan Arc. (B) Synoptic calendar and positions of great/large earthquakes along the Himalayan front (including





instrumental, historical and paleoseismic events). Orange horizontal bars approximate
minimum source lengths with or without observed surface rupture. Vertical blue bars
correspond to the radiocarbon-model constraints on the timing of the different events. Vertical
brown bars correspond to ~2600-yr-long record deduced from the present study.





| Unit | Sample name | Nature | Measured radiocarbon age (years B.P.) | Calibrated ages (Calendric, 2σ) | C [mg] | δ¹³C value |
|---|---|---|---|---|---|---|
| W1 | PI-C24 | bark | 140.6 ± 0.44 pMC | modern | 1.20 | -28.8 |
| W1 | PI-C23 | bark | 118.29 ± 0.31 pMC | modern | 3.60 | -25.9 |
| U2 | PI-C43 | charcoal | 1520 ± 30 | AD 520 ± 95 | 1.08 | -24.7 |
| U2 | PI-C35 | charcoal | 1770 ± 30 | AD 330 ± 60 | 0.39 | -33.1 |
| *U2* | *PI-C33* | *charcoal* | *2405 ± 30* | *BC 565 ± 160* | *1.00* | *-29.1* |
| U3 | PI-C37 | charcoal | 1730 ± 30 | AD 240 ± 100 | 1.77 | -30.3 |
| U3 | PI-C40 | charcoal | 1960 ± 30 | AD 45 ± 85 | 0.87 | -27.9 |
| *U3* | *PI-C38* | *charcoal* | *2560 ± 30* | *BC 680 ± 125* | *0.77* | *-26.1* |
| U5 | PI-C09 | charcoal | 2180 ± 30 | BC 270 ± 100 | 1.26 | -27.6 |
| U5 | PI-C19 | charcoal | 2240 ± 30 | BC 300 ± 95 | 2.62 | -31.3 |
| U5 | PI-C28 | charcoal | 2285 ± 30 | BC 310 ± 100 | 2.01 | -31.8 |
| U5 | PI-C16 | charcoal | 2280 ± 30 | BC 310 ± 100 | 1.91 | -30.6 |
| *U5* | *PI-C11* | *charcoal* | *2905 ± 30* | *BC 1110 ± 100* | *0.89* | *-28.8* |
| *U5* | *PI-C12* | *charcoal* | *2860 ± 30* | *BC 1025 ± 95* | *0.86* | *-28.1* |
| U6 | PI-C06 | charcoal | 2495 ± 30 | BC 660 ± 125 | 1.87 | -26.3 |
| U6 | PI-C05 | charcoal | 2485 ± 35 | BC 645 ± 140 | 1.90 | -29.6 |
| U6 | PI-C36 | charcoal | 2510 ± 30 | BC 665 ± 125 | 2.55 | -22.7 |
| U6 | PI-C42 | charcoal | 2480 ± 30 | BC 645 ± 135 | 1.67 | -20.8 |
| U6 | PI-C44 | charcoal | 2590 ± 30 | BC 710 ± 115 | 1.04 | -32.1 |
| U6 | PI-C48 | charcoal | 2545 ± 30 | BC 675 ± 130 | 1.17 | -27.5 |
| U6 | PI-C29 | charcoal | 2625 ± 30 | BC 805 ± 30 | 1.26 | -26.8 |
| *U6* | *PI-C46* | *charcoal* | *37700 ± 800* | *BC 40080 ± 1280* | *0.59* | *-28.4* |

[a]See trench log for stratigraphic unit designations.

[b]Radiocarbon years B.P. relative to 1950 A.D. (with 1 σ counting error). All samples have been dated by the Poznan Radiocarbon Laboratory.

[c]Calendric dates were calibrated using OxCal and the atmospheric calibration curve IntCal13. Calendric ages have been rounded to the nearest ½ decade assuming the 5 years accuracy of the IntCal13 curve. D Calendric dates were calibrated using the atmospheric calibration curve IntCal09 for the Northern Hemisphere



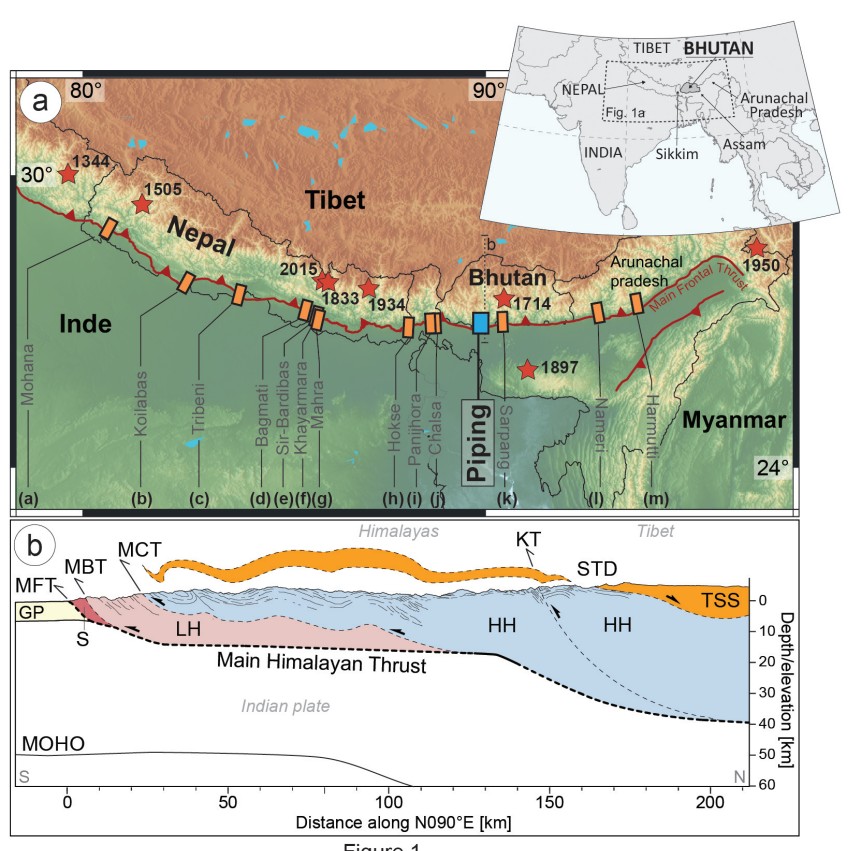

Figure 1



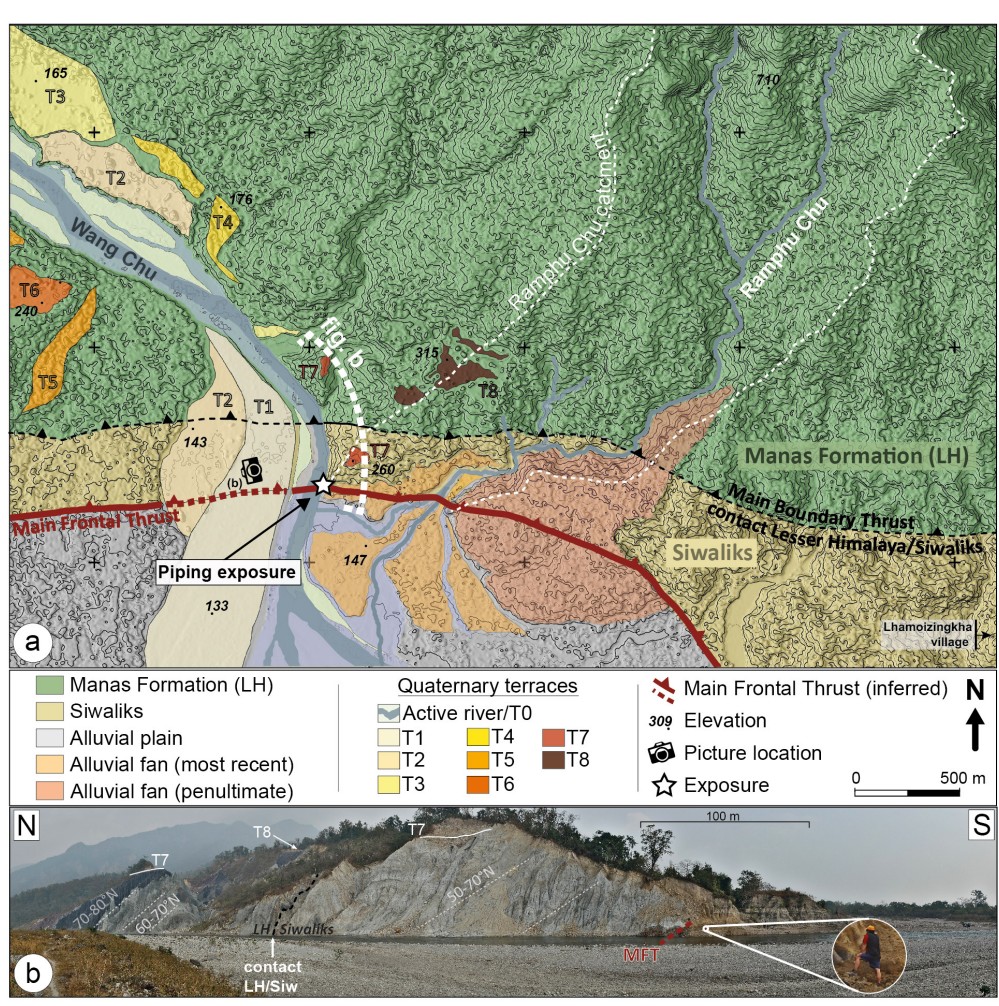

Figure 2





Figure 3



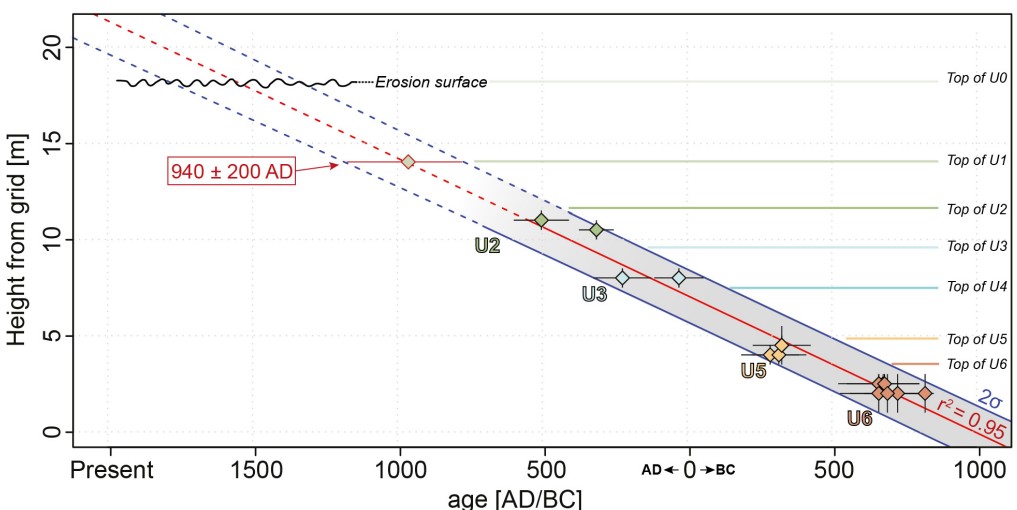

Figure 4



Figure 5



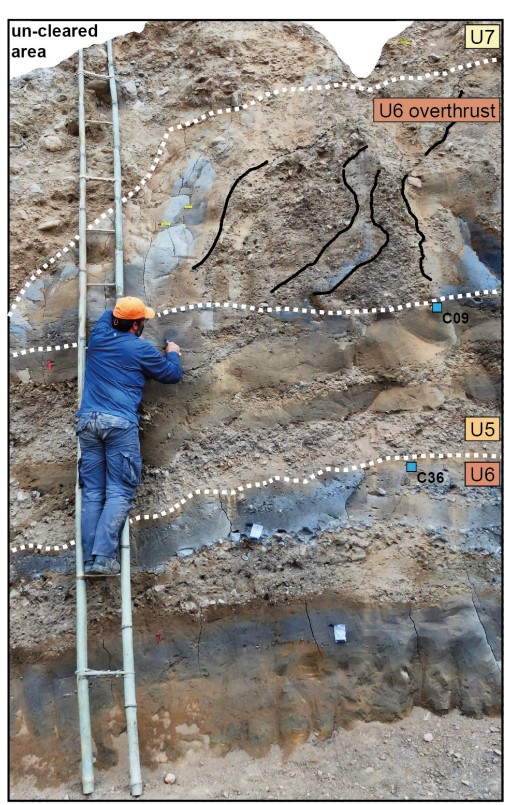

Figure 6





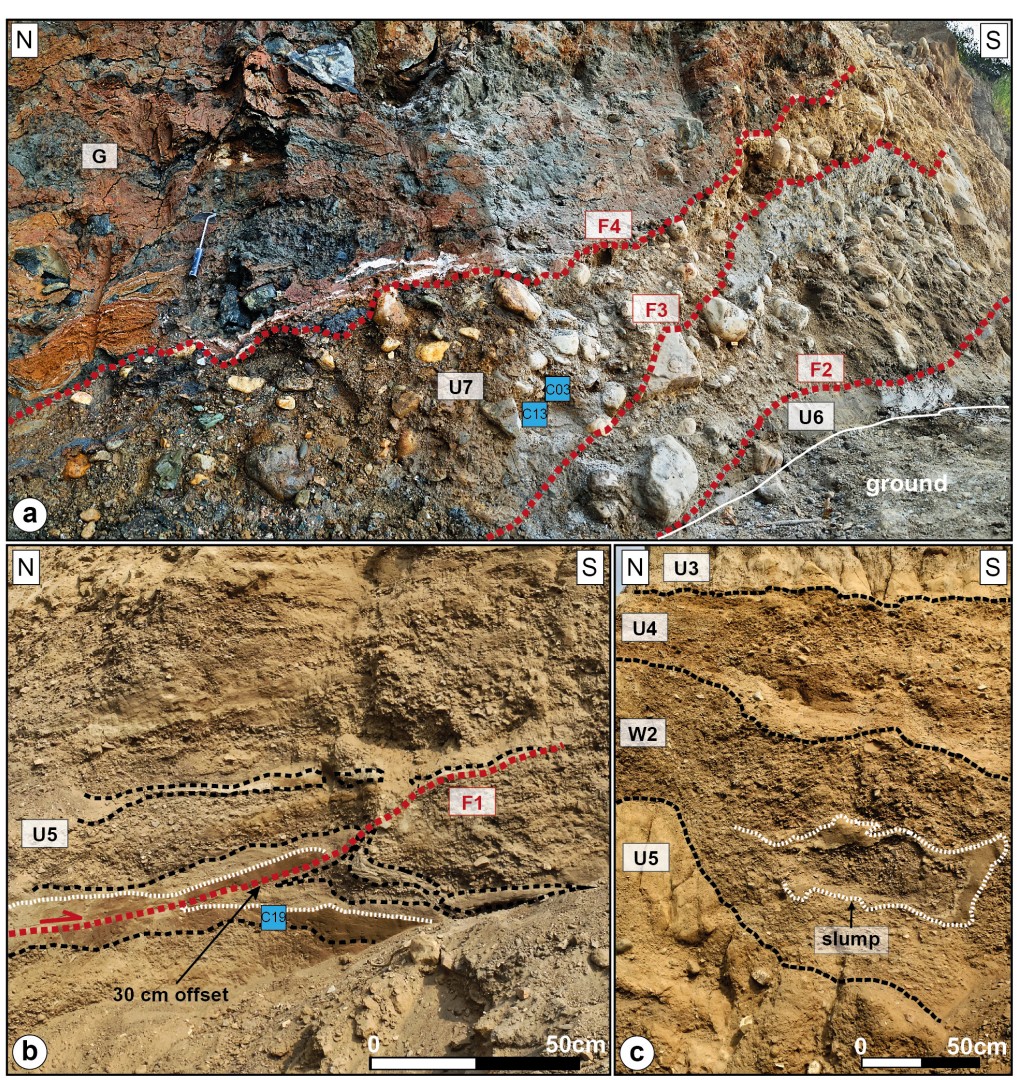

Figure 7

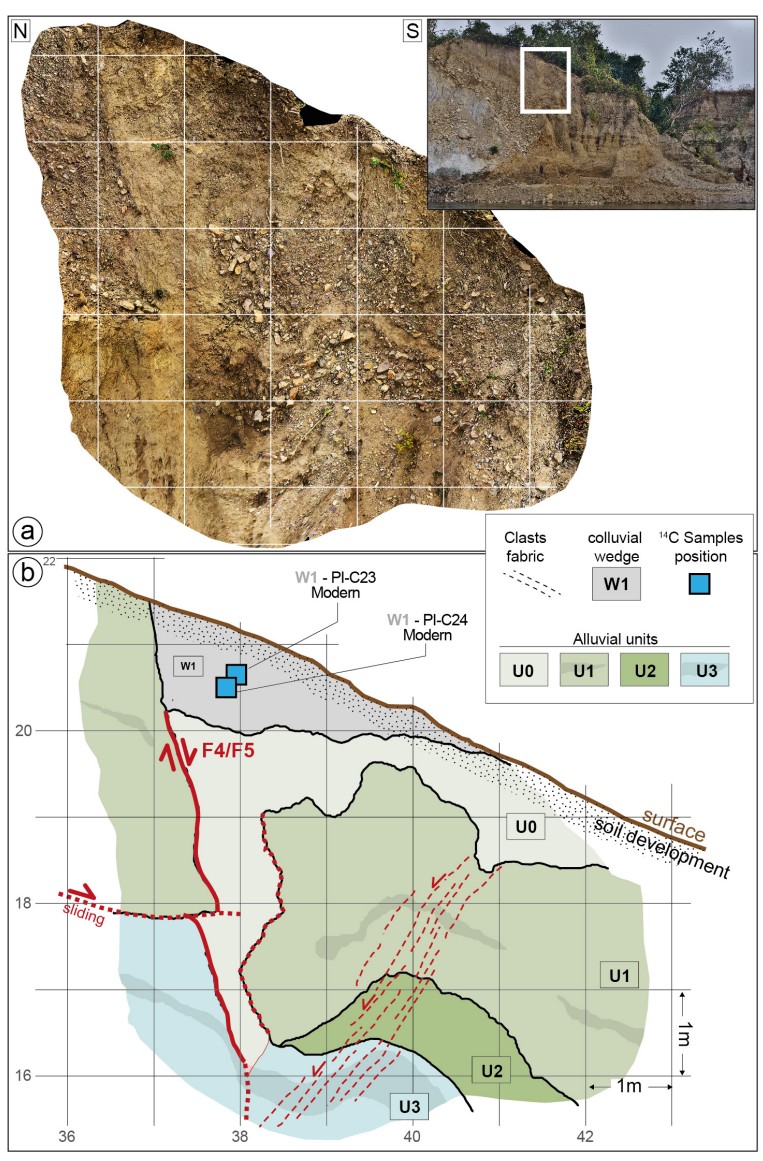

Figure 8



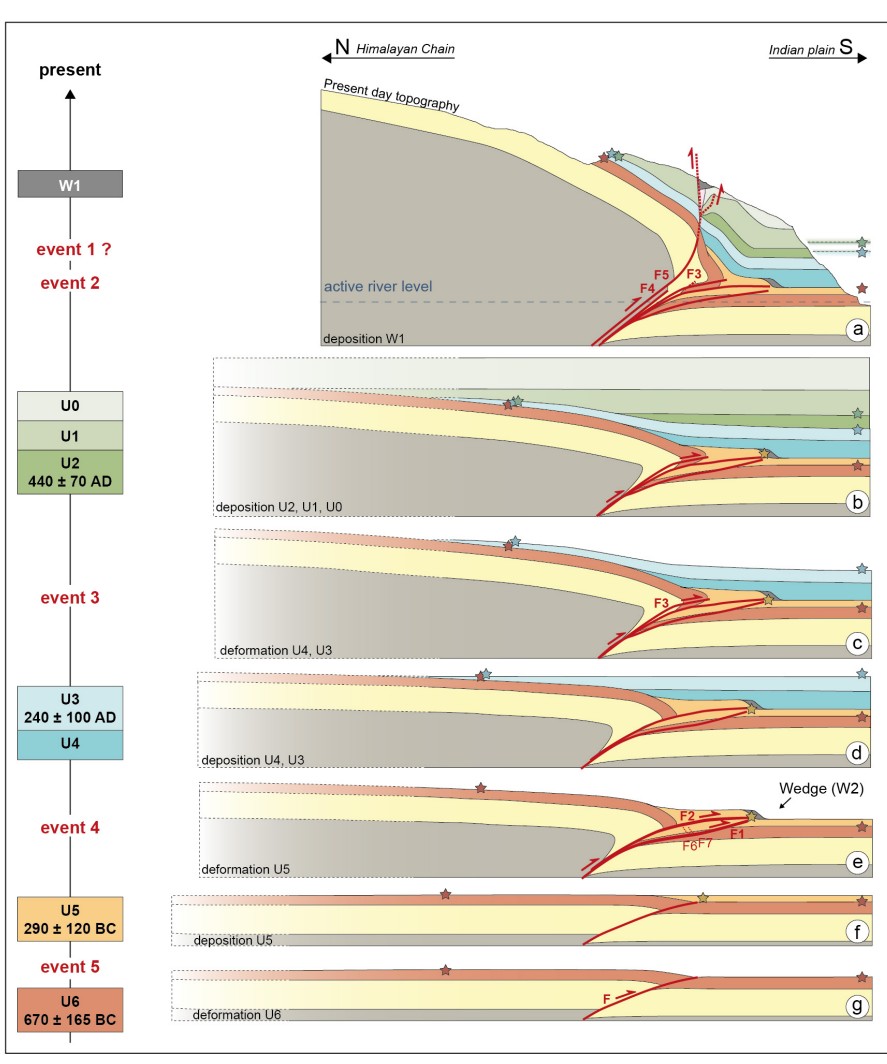

Figure 9



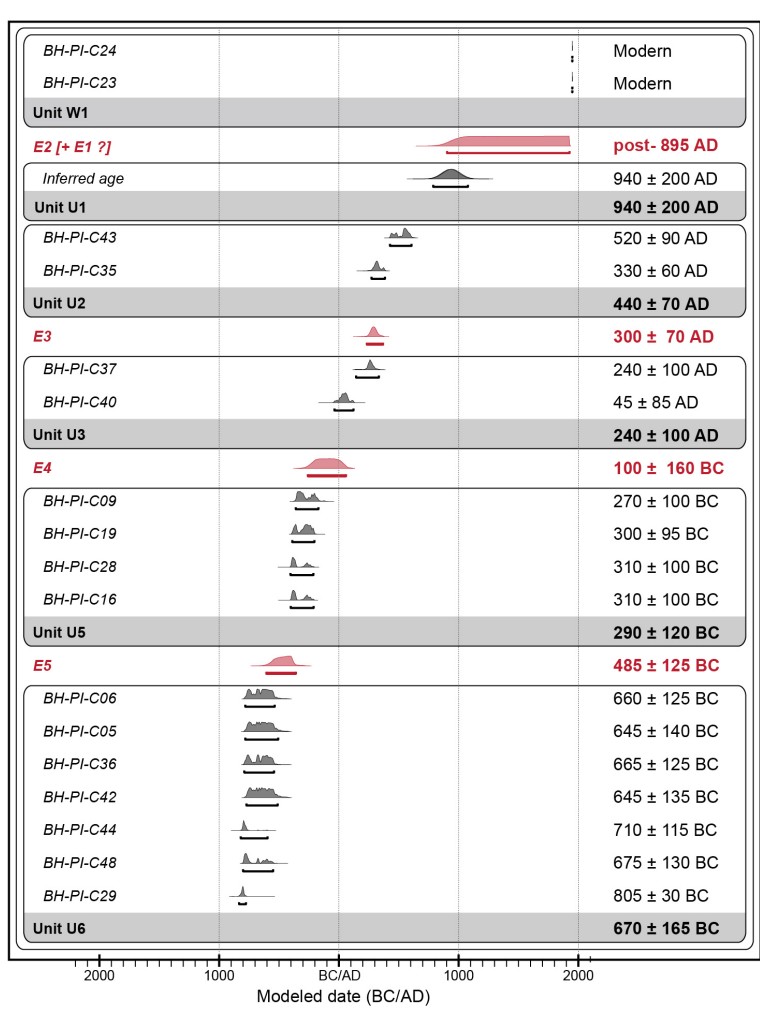

Figure 10



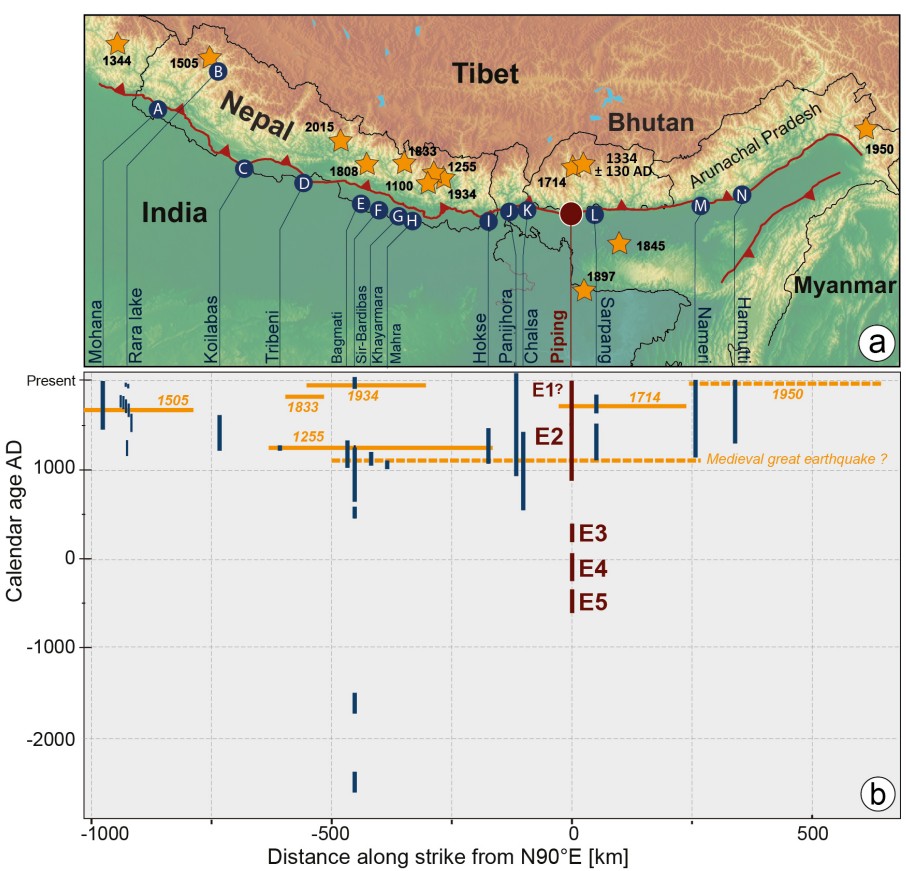

Figure 11