# Peer review of "A 2600-yr-long paleoseismic record for the Himalayan Main Frontal Thrust"

_Solid Earth, 2020_

## Referee Comment (RC1) · Anonymous Referee #1 · 2 Jun 2020

The manuscript presents some interesting results on the timing and recurrence intervals of the past earthquakes from Bhutan, Central Himalaya. The main contribution and originality of the study lie in the detailed paleoseismic investigation of a river cut section along the Main Frontal Thrust (MFT) to reconstruct the past earthquake chronology and the average slip rate. The authors have suggested five events and a slip rate of 25.3±4 mm/yr along the Piping Site. The MFT is one of the major plate convergence thrust fault accommodating substantial slip, the estimation of earthquake size, time, and recurrence interval is valuable towards the seismic hazard assessment. Thus I, think the manuscript is indeed important and would draw interest from a wide range of earth scientists and fits the scope of Solid Earth. The manuscript is well written

and organized. However, there are some inaccuracies with the interpretation which is discussed in the following section. I encourage the editor to request for a thoughtful revision to address these issues which will refine the manuscript before it is published.

Specific Comments:

Line 45: Comparing Strike-slip Fault and Thrust Fault systems does not seem probable. Kindly quote examples from thrust fault setting.

Line 102: Kindly mention the classification of terraces (youngest-oldest) for easy understanding.

Line 103: The author keeps switching between Siwalik and Siwaliks. In general, Siwaliks are used when you want to include Upper and Lower Siwalik, else use Siwalik.

Line 105: Remove 's' from 'covers'.

Line 107: Instead of 'low relative elevation' you can use 'relatively lower elevations'.

Line 259-261: Not clear. Please break it or re-write it.

Line 264: Because of the low resolution of the image file and non-uniform cleaning of the wall it is difficult to judge the deformation caused by F1. I recommend the author to add a blow-up image of the fault structure from grid 40-45.

Line 266: Along F6 and F5, no strong deformation is visible, instead U6 is rather horizontal with minor undulation which is quite common with fluvial deposits. Kindly check the validity of both the faults.

Line 269: In Fig. 6 what does the white dashed lines represent? Kindly mark and classify the faults in the figure. Also extend and mark F2 in Fig. 7b.

Line 281: With the present resolution of Fig. 5 it is difficult to see F5. Kindly mark F5 in Fig. 7a.

Line 313: I don't understand how retro-deformation of U0 and U2 is carried out and

compared with U1 because these two units are absent in the hanging-wall and the thickness may have varied. Thus, distinguishing E1 and E2 becomes difficult.

Line 325: Kindly clarify if 1.5 m is vertical offset or coseismic slip.

Line 335-336: How the slip along F4 is calculated using the dip of F3? The F3 has not deformed or cross-cut U3 and U4 which means they were deposited after the event on F3. The inference that the slip propagated along F3 seems improbable instead it looks it occurred along F4 and F5.

Line 347: It is not fully clear why U5 is recording ∼13.5 m of dip-slip whereas U6 is recording only ∼9 m of dip-slip. In fact, it should be the other way around as U6 is an older unit and should record more slip if there were two events.

Line 352: The author gives a detailed description of retro-deformation i.e., extracting elevation from the highest observable point of each unit to their corresponding point in the undeformed section in the footwall. However, my major concern lies in how the retro-deformation of U5 is carried out to evaluate two events. Either by restoring dip-slip of U5 along F2 or by restoring its highest point which is around the grid (38,12). With the current explanation and no visible cross-cutting structure in the section, it is difficult to distinguish E4 & E5.

Line: 414: It seems unlikely for an earthquake reaching magnitude 8.5 to have a slip of only 1.5 m. There is some evidence of earthquakes from Himalaya recording lesser slip for example Mw 7.8 1905 Kangra earthquake but there was slip partitioning reported between thrust and strike-slip faults (Szeliga and Bilham, 2017).

Please also note the supplement to this comment:
https://www.solid-earth-discuss.net/se-2020-59/se-2020-59-RC1-supplement.pdf

---

## Referee Comment (RC2) · Anonymous Referee #2 · 23 Jun 2020

%%%%%%%General comments%%%%%%%

I enjoyed reading this manuscript that bring new constraints on the timing of paleo-earthquakes and the return period of surface rupturing events along the Main Frontal Thrust in Piping, western Bhoutan. Indeed, this study documents an exceptional natural rivercut of the frontal thrust at the junction between two river catchments. The active fault affects the alluvial cone of the tributary of the main river while a flight of alluvial terraces were abandoned and preserved along the main stream. The work done at the front of this outcrop is spectacular and well documented. The text reads well and is informative. Most of the conclusions appear supported by the observations

and are documented in the text. However, some observations appear to me either less convincing given their present documentation or they should be associated with less weight in the conclusion. I am personally not confident with the conclusion regarding the number of earthquakes. I further find that the estimate of the cumulated slip on the fault is not associated with reasonable uncertainties. I finally regret the absence of confrontation between the paleoearthquake ruptures and the alluvial terrace abandonment in the hangingwall of the thrust. My conclusion is therefore that the article's need moderate revisions including additional arguments or that the conclusions need some slight down tuning.

%%%%%%%%% Specific comments: %%%%%%%%%

I have not been convinced by the documentation of event E1 which affects the same units as E2. I cannot make the difference between one and the other and recommend either to find complementary observations or argument for the existence of two earthquakes at this site. Without any additional arguments on the outcrop I would personally go for documenting a scenario with one earthquake E1 (unless finding two generations of abandoned terraces with ages consistent with the two earthquakes), further mentioning that two earthquakes were described at sites further east within the period post 940AD, suggesting there is possibly more than one. I had a really hard time in understanding how the retro-deformation constrained so precisely the cumulated slip on the fault ($\sim$40.2 m over 1629+/-255 yr line 418 !) given the significant uncertainty on the dip of the faults associated to the fact that most of the observations control mainly the vertical offsets which needs to be translated in slip on the fault. An estimate of the uncertainties associated to the estimate of the total amount of slip could be more realist. I regret that the manuscript does not integrate at least a paragraph on the relations between paleo-earthquakes and alluvial terraces that might have been abandoned after tectonic events. Indeed, some of the events described in the manuscript accommodate enough to create more than 10 meters of vertical offsets and might have been followed by episodes of severe incision. I do not understand the reason why these terraces,

were not dated in order to facilitate both the determination of the incremental incision. I suggest that section 4 incorporates a discussion on eventual relations between events and terrace abandonment.

%%%%%%%%%% Additional remarks on text and figures:%%%%%%

Line 22-25 : Far field convergence is estimated at 17.0+/-0.5 mm/yr ( Marechal et al., 2016 ). The average slip rate @ "25.3+/-4 mm/yr" is therefore significantly larger than the geodetic and geomorphological results. Line 82-83 "This last event contributes to the debate about the possible deficit of seismic moment …..". I do not understand, please, rephrase. Line 107: replace at low relative elevation by " at low elevation above the present day river course". Line 223 : Wedge W1 is described as affected by intense internal deformation. Is it true ? If true, it should be documented on Figure 8 Line 390: Need to downtune the paragraph on E1, provided that there are no additional informations than those described here. Indeed, this earthquake is not documented properly at this site and the constraints appear elusive to me.

Figure 1: Damak trench is not on the map (Wesnousky et al., 2016) nor Charnath trench (Rizza et al., 2019). Bagmati, Sir Bardibas, Kayarmara and Mahara are not at the right place.

Figure 2: I suggest reporting a few altitudes along the Wang Chu course (as well as in terrace T2 North and T5 so that the reader can estimate the elevation of the terraces above the river without going back to the text. I am surprised to see that the supposed trace of the Main Boundary Thrust and Main Frontal Thrust are straight through the river, without drawing a small V in the valley toward the North, a likely feature despite the relatively steep fault dips. I recommend to check properly the shape of the trace of the fault provided it dips with the value mentionned in the text.

Figure 8: The wedge W1 is given in the text as intensely deformed ("… exhibit little stratigraphy, intense internal deformation") Line 223. Would it be possible to see that on this figure ? or with a zoom on W1 in supplementary data material ?

Figure 9: Scale is needed. The amount shortening or coseismic slip should be reported on this figure at every step, with their uncertainty.

Figure 10: Add unit W2

---

## Author Comment (AC1) · 5 Aug 2020

Specific Comments: Line 45: Comparing Strike-slip Fault and Thrust Fault systems does not seem probable. Kindly quote examples from thrust fault setting. -> We disagree. We are not comparing kinematics, only the length of paleoseismic records with respect to the length of the structure for regional-scale faults capable of producing major to great earthquakes.

Line 102: Kindly mention the classification of terraces (youngest-oldest) for easy understanding. -> We agree. Action: added "(younger)" and "(older)" after height.

Line 103: The author keeps switching between Siwalik and Siwaliks. In general, Siwaliks are used when you want to include Upper and Lower Siwalik, else use Siwalik. -> We agree. Action: We checked and corrected throughout the text.

Line 105: Remove 's' from 'covers'. -> We agree. Action: we implemented proposed corrections.

Line 107: Instead of 'low relative elevation' you can use 'relatively lower elevations'. -> We agree. Action: for clarity, we removed "relative" and added "above present stream".

Line 259-261: Not clear. Please break it or re-write it. -> We agree. Action: we re-wrote the sentence for easier reading.

Line 264: Because of the low resolution of the image file and non-uniform cleaning of the wall it is difficult to judge the deformation caused by F1. I recommend the author to add a blow-up image of the fault structure from grid 40-45. -> We agree. Action: we added full-resolution orthomosaics as Supplementary Material.

Line 266: Along F6 and F5, no strong deformation is visible, instead U6 is rather horizontal with minor undulation which is quite common with fluvial deposits. Kindly check the validity of both the faults. -> We disagree. F5 is not mentioned in that line, but F6 and F7. Localized displacements reach several 10s of cm, which is significant but may appear weak compared to the scale of the exposure. The full-resolution orthomosaic (previous comment) and added blow up in Fig. 7d should clarify this point.

Line 269: In Fig. 6 what does the white dashed lines represent? Kindly mark and classify the faults in the figure. Also extend and mark F2 in Fig. 7b. -> We agree. White lines in Figure 6 are confusing. Action: we modified Figure 6, better defined the various lines in the caption and extended F2 in Figure 7b and 7c.

Line 281: With the present resolution of Fig. 5 it is difficult to see F5. Kindly mark F5 in Fig. 7a. -> We agree. Action: we modified Figure 7a to better show F5.

Line 313: I don't understand how retro-deformation of U0 and U2 is carried out and
compared with U1 because these two units are absent in the hanging-wall and the thickness may have varied. Thus, distinguishing E1 and E2 becomes difficult. -> We agree. Action: In agreement with suggestions from both Reviewers, we combined E1 and E2 into a single event with the possibility for a supplementary one.

Line 325: Kindly clarify if 1.5 m is vertical offset or coseismic slip. -> We agree. Action: we replaced "co-seismic slip of  ${\sim}1.5$  m" with "vertical component of co-seismic slip of  ${\sim}1.5$  m".

Line 335-336: How the slip along F4 is calculated using the dip of F3? The F3 has not deformed or cross-cut U3 and U4 which means they were deposited after the event on F3. The inference that the slip propagated along F3 seems improbable instead it looks it occurred along F4 and F5. -> We agree. This is a typo; at line 336 "slip for E3 along F4" should read "slip for E3 along F3". Action: we corrected the typo.

Line 347: It is not fully clear why U5 is recording 13.5 m of dip-slip whereas U6 is recording only 9 m of dip-slip. In fact, it should be the other way around as U6 is an older unit and should record more slip if there were two events. -> We disagree. These values are estimates of co-seismic slip, not cumulative.

Line 352: The author gives a detailed description of retro-deformation i.e., extracting elevation from the highest observable point of each unit to their corresponding point in the undeformed section in the footwall. However, my major concern lies in how the retro-deformation of U5 is carried out to evaluate two events. Either by restoring dip slip of U5 along F2 or by restoring its highest point which is around the grid (38,12). With the current explanation and no visible cross-cutting structure in the section, it is difficult to distinguish E4 & E5. -> We disagree. Grid point (38,12) corresponds to the highest point of U5 and has recorded cumulative deformation, whereas our retro-deformation consists in sliding along F2 to determine the co-seismic slip associated with event E4. Furthermore, the characterization of E5 is solely based on the geometry of U6; at this point U5 does not exist yet.
Line: 414: It seems unlikely for an earthquake reaching magnitude 8.5 to have a slip of only 1.5 m. There is some evidence of earthquakes from Himalaya recording lesser slip for example Mw 7.8 1905 Kangra earthquake but there was slip partitioning reported between thrust and strike-slip faults (Szeliga and Bilham, 2017). -> We disagree. Hétényi et al. (2016) conclude the AD 1714 earthquake may have reached Mw 7.5-8.5, not 8.5 and we accurately quoted their results. We do not propose a magnitude ourselves. Furthermore, there are very few observed ruptures to compare to (slip partitioning certainly rules out the Kangra earthquake as a proper comparison as pointed out by the Reviewer).

Please also note the supplement to this comment: https://se.copernicus.org/preprints/se-2020-59/se-2020-59-AC1-supplement.pdf

---

## Author Comment (AC2) · 5 Aug 2020

Specific comments: I have not been convinced by the documentation of event E1 which affects the same units as E2. I cannot make the difference between one and the other and recommend either to find complementary observations or argument for the existence of two earthquakes at this site. -> We agree. Action: see next comment.

Without any additional arguments on the outcrop I would personally go for documenting a scenario with one earthquake E1 (unless finding two generations of abandoned terraces with ages consistent with the two earthquakes), further mentioning that two earthquakes were described at sites further east within the period post 940AD, sug-

gesting there is possibly more than one. -> We agree. We thank the Reviewer for this helpful suggestion and modify the text (sections 3.3 and 4) accordingly.

I had a really hard time in understanding how the retro-deformation constrained so precisely the cumulated slip on the fault (40.2 m over 1629+/-255 yr line 418 !) given the significant uncertainty on the dip of the faults associated to the fact that most of the observations control mainly the vertical offsets which needs to be translated in slip on the fault. An estimate of the uncertainties associated to the estimate of the total amount of slip could be more realist. -> We agree. Although we do not mention "40.2 m" but "∼40.2 m", we agree a more detailed assessment of uncertainties should be provided. Action: we added details on uncertainties on slip measurements, dip angles and inferred slip values.

I regret that the manuscript does not integrate at least a paragraph on the relations between paleo-earthquakes and alluvial terraces that might have been abandoned after tectonic events. Indeed, some of the events described in the manuscript accommodate enough to create more than 10 meters of vertical offsets and might have been followed by episodes of severe incision. -> We disagree. The distribution of terrace heights (∼1 m, ∼11 m, ∼33 m, ∼43 m, ∼80 m, ∼90 m, ∼100 m and ∼170 m) does not suggest clear systematic abandonment associated with repeated co-seismic uplift. Including them in our analysis without evidence for faulting and adequate determination of faulting ages would be model-driven.

I do not understand the reason why these terraces, were not dated in order to facilitate both the determination of the incremental incision. I suggest that section 4 incorporates a discussion on eventual relations between events and terrace abandonment. -> We disagree. Most terraces at this site were dated and we have accumulated a significant dataset on incision rates. However, presenting said dataset is a whole different study that will be written up and submitted at a later stage.

Additional remarks on text and figures: Line 22-25 : Far field convergence is estimated

at 17.0+/-0.5 mm/yr ( Marechal et al., 2016 ). The average slip rate @ "25.3+/-4 mm/yr" is therefore significantly larger than the geodetic and geomorphological results. -> After implementation of a detailed analysis of uncertainties, our final slip rate is 24.9 $\pm$ 10.4 mm/yr, which largely encompasses geodetic and geomorphological results. Action: none.

Line 82-83 "This last event contributes to the debate about the possible deficit of seismic moment . . ...". I do not understand, please, rephrase. -> We agree. Our wording was not clear enough. Action: we rewrote that sentence.

Line 107: replace at low relative elevation by " at low elevation above the present day river course". -> We agree. Action: we implemented proposed modification.

Line 223 : Wedge W1 is described as affected by intense internal deformation. Is it true ? If true, it should be documented on Figure 8. -> We agree. This is a mistake, only W2 exhibits intense internal deformation. Action: we corrected the text accordingly.

Line 390: Need to downtune the paragraph on E1, provided that there are no additional informations than those described here. Indeed, this earthquake is not documented properly at this site and the constraints appear elusive to me. -> We agree. Action: see previous comments that address this point in detail.

Figure 1: Damak trench is not on the map (Wesnousky et al., 2016) nor Charnath trench (Rizza et al., 2019). Bagmati, Sir Bardibas, Kayarmara and Mahara are not at the right place. -> We agree. Action: we corrected Figure 1 following suggestions from R2.

Figure 2: I suggest reporting a few altitudes along the Wang Chu course (as well as in terrace T2 North and T5 so that the reader can estimate the elevation of the terraces above the river without going back to the text. -> We agree. Action: we modified Figure 2 following suggestions from R2.

I am surprised to see that the supposed trace of the Main Boundary Thrust and Main

Frontal Thrust are straight through the river, without drawing a small V in the valley toward the North, a likely feature despite the relatively steep fault dips. I recommend to check properly the shape of the trace of the fault provided it dips with the value mentionned in the text. -> We disagree. The MBT is very steep here (as shown in Fig. 2b) and the section of MFT mapped does not affect strong reliefs; alluvial fans exhibit shallow slopes and the floor of the Wang Chu plain is basically flat. There is no geometric reason to invoque a V-shaped trace.

Figure 8: The wedge W1 is given in the text as intensely deformed (". . . exhibit little stratigraphy, intense internal deformation") Line 223. Would it be possible to see that on this figure ? or with a zoom on W1 in supplementary data material ? -> We agree. As addressed in this Reviewer's comment on Line 223. Action: see comment on Line 223.

Figure 9: Scale is needed. The amount shortening or coseismic slip should be reported on this figure at every step, with their uncertainty. -> We disagree. Figure 9 is a schematic illustration of the retro-deformation process. Our analysis of co-seismic displacements is based on the direct analysis of the original log and inferences stated in the text. Thus, adding co-seismic values on Figure 9 would suggest said values were measured from the Figure and be misleading to the reader.

Figure 10: Add unit W2 ->We disagree. This is a chronostratigraphic model and there are no samples collected from W2 to be displayed here. W2 being a scarp-derived slump from unit U5, its age would be that of U5, anyway. The main text as well as Figure 9 do explain the place of W2 within the stratigraphic section and its relationship to event E4.

Please also note the supplement to this comment:
https://se.copernicus.org/preprints/se-2020-59/se-2020-59-AC2-supplement.pdf
* * *
[Figure]

**Fig. 1.** Figure 1 corrected following suggestions from reviewer

[Figure]

Figure 2

**Fig. 2.** Figure 2 corrected following suggestions from reviewer

---

## Author Response (AR2)

Black text: reviewer comment
Green text: Author answer

**Topical Editor Decision**

I agree with reviewer 2 that there should be an approximate scale given in Figure 9.
We agree.
Action: added approximative scale in Figure 9.

Reviewer 1 comment about line 347 - please add a clarification to the text that these values are coseismic slip to avoid any confusion.
We agree.
Action: We added the "co-seismic" to the sentence to avoid any confusion.
*"Hence, restoring U5 involves removing W2 then retro-sliding the duplexed part of U5 along F2 to bring grid point (51.5, 5) back to its minimal original position at grid point (39.5, 4) with a __co-seismic__ dip-slip offset of 13.5 m ± 0.6 m along F2."*